# Detection of Tumour-Targeted IRDye800CW Tracer with Commercially Available Laparoscopic Surgical Systems

**DOI:** 10.3390/diagnostics13091591

**Published:** 2023-04-29

**Authors:** Daan J. Sikkenk, Andrea J. Sterkenburg, Iris Schmidt, Dimitris Gorpas, Wouter B. Nagengast, Esther C. J. Consten

**Affiliations:** 1Department of Surgery, University Medical Centre Groningen, University of Groningen, Hanzeplein 1, 9713 GZ Groningen, The Netherlands; 2Department of Surgery, Meander Medical Centre, Maatweg 3, 3813 TZ Amersfoort, The Netherlands; 3Department of Gastroenterology, University Medical Centre Groningen, University of Groningen, Hanzeplein 1, 9713 GZ Groningen, The Netherlands; 4Institute of Biological and Medical Imaging, Helmholtz Zentrum München (GmbH), Ingolstädter Landstraße 1, D-85764 Neuherberg, Germany; 5Chair of Biological Imaging, Center for Translational Cancer Research (TranslaTUM), Technical University of Munich, Ismaninger Straße 22, D-81675 Munich, Germany

**Keywords:** fluorescence guided surgery, fluorescence molecular imaging, image-guided surgery, near-infrared fluorescence (NIRF), targeted tracer, IRDye800CW, indocyanine green (ICG)

## Abstract

(1) Introduction: Near-infrared fluorescence (NIRF) combined with tumour-targeted tracers, such as bevacizumab-800CW, could aid surgical decision-making. This study explored the use of IRDye800CW, conjugated to bevacizumab, with four commercially available NIRF laparoscopes optimised for indocyanine green (ICG). (2) Methods: A (lymph node) phantom was made from a calibration device for NIRF and tissue-mimicking material. Serial dilutions of bevacizumab-800CW were made and ICG functioned as a reference. System settings, working distance, and thickness of tissue-mimicking material were varied to assess visibility of the fluorescence signal and tissue penetration. Tests were performed with four laparoscopes: VISERA ELITE II, Olympus; IMAGE1 S™ 4U Rubina, KARL STORZ; ENDOCAM Logic 4K platform, Richard Wolf; da Vinci Xi, Intuitive Surgical. (3) Results: The lowest visible bevacizumab-800CW concentration ranged between 13–850 nM (8–512 times diluted stock solution) for all laparoscopes, but the tracer was not visible through 0.8 cm of tissue in all systems. In contrast, ICG was still visible at a concentration of 0.4 nM (16,384 times diluted) and through 1.6–2.4 cm of tissue. Visibility and tissue penetration generally improved with a reduced working distance and manually adjusted system settings. (4) Conclusion: Depending on the application, bevacizumab-800CW might be sufficiently visible with current laparoscopes, but optimisation would widen applicability of tumour-targeted IRDye800CW tracers.

## 1. Introduction

Surgeons rely on visual and tactile feedback for intraoperative decision-making. However, many laparoscopic or robotic surgical systems lack tactile feedback. Image-guided surgery, such as near-infrared fluorescence (NIRF), aid surgical decision-making with additional visual feedback. There has been an increasing interest in NIRF and its new applications for fluorescence guided surgery. In oncological surgery, NIRF is primarily used to achieve tumour-free excision, identify occult lesions, debulking, and recognition of vital structures [1].

Near-infrared light (650–950 nm for the first NIRF window) has superior tissue penetration compared to white light and has limited interference with human tissues [2,3,4,5]. Fluorescence-guided surgery is primarily focused on the non-specific tracer indocyanine green (ICG) as it has a well-established safety profile and is the only FDA and EMA approved fluorophore for general surgery [6]. Development of new fluorescent tracers is mostly focused on tumour-targeted tracers; fluorophores bound to a ligand [7,8]. Tumour-targeted tracers could provide real-time information about the presence of tumour or metastases and allow intraoperative decision-making [9]. Furthermore, antibodies are commonly used as a ligand due to their availability for a wide range of targets and modifiability [10].

The fluorophore IRDye800CW has an NH ester group for labelling proteins such as antibodies [11], making it an applicable fluorophore for tumour-targeted tracers. By using the same fluorophore for multiple antibodies, the same camera system could be used for a wide variety of tumour-targeted tracers and indications. Multiple commercially available open-air NIRF camera systems can detect IRDye800CW (absorption: ~780 nm, excitation: ~800 nm [11,12]), but are primarily intended for ICG (absorption: ~800 nm, excitation: ~830 nm [13]) and cannot be used for minimally invasive surgery [14]. Bevacizumab-800CW is a tumour-targeted tracer, aimed at vascular endothelial growth factor α (VEGFα), which has been used in multiple clinical studies [9,15,16,17,18,19,20,21,22]. Likewise, cetuximab-800CW, aimed at endothelial growth factor, is used in multiple clinical trials [23,24], and various other antibodies conjugated to IRDye800CW are currently investigated [25,26,27] since the process of developing these tracers is relatively simple compared to developing a new investigational drug or tracer [27,28]. Micro-dosages of tumour-targeted tracers are most often used in studies since first-in-patient studies can more easily be conducted due to the unlikelihood of major side effects [29,30,31,32]. Simultaneously, tumour-targeted tracers in micro-dosages still provide sufficient fluorescent intensity for diagnostic purposes.

Our intention is to use targeted tracers for the detection of sentinel lymph nodes or lymph node metastases in patients with colon cancer during laparoscopic surgery [33,34]. Sentinel lymph node detection entails interstitial injection of a tracer, which results in a local high concentration of the tracer. However, there is a limited number of studies regarding commercially available laparoscopes for use with (tumour-targeted) IRDye800CW tracers [35,36]. Comparison of different systems is difficult as intraoperative detection of a fluorophore is dependent on various biological and optical factors [37]. Therefore, we conducted a systematic in vitro study with a NIRF calibration device. This phantom study aimed to explore the use of IRDye800CW, as an example conjugated to bevacizumab, with four commercially available NIRF laparoscopic surgical systems.

## 2. Materials and Methods

### 2.1. Phantom Set-Up

Clinical-grade bevacizumab-800CW was manufactured according to current Good Manufacturing Practice and supplied by the University Medical Center Groningen for use in this phantom study, as previously described [28]. In summary, bevacizumab (Roche Holding AG, Basel, Switzerland) and IRDye-800CW-NHS (LI-COR Biosciences, Lincoln, NE, USA) were conjugated with a dye to antibody ratio of 1.6:1 and formulated in a sodium chloride solution. Fifteen concentrations were created using serial dilution, starting from the standard concentration 6.8 µM (1 mg/mL bevacizumab-800CW), using 1:1 dilution with PBS, ending at 0.42 nM.

ICG (Verdye, Diagnostic Green GmbH, Aschheim-Dornach, Germany) was used as a fluorescent reference for bevacizumab-800CW. ICG was combined with 2% intralipid and PBS. Intralipid acted as human albumin in blood serum and was required for this phantom study, as ICG dissolved in water for injection or PBS has a different excitation and emission wavelength [38]. The ICG solution was serially diluted from the manufacture’s recommended dose (5 mg/mL = 6.5 mM), similar to bevacizumab-800CW, resulting in the lowest concentration of 0.39 µM.

The molar concentration of ICG serial dilutions were a factor 950 higher than the bevacizumab-800CW concentrations, and 597 times higher than the molar concentration IRDye800CW due to the 1.6:1 dye to antibody ratio. Therefore, the first five concentrations of bevacizumab-800CW serial dilutions correspond approximately to the last five concentrations of ICG (Appendix A Table A1).

Transparent polypropylene Eppendorf tubes were filled with the solutions of bevacizumab-800CW or ICG and placed in a CalibrationDisk (SurgVision, Munich, Germany) [37,39]. The CalibrationDisk is a calibration device for NIRF systems and can hold up to eight Eppendorf tubes (Figure 1a). Two windows in the calibration device, approximately the size of lymph nodes, allowed fluorescent intensity measurement for each tube. Fat tissue-mimicking hydrogel made from gelatine (2.5 g), agar (2.5 g), and water (200 mL) to recreate fat in the mesocolon, were used to cover the CalibrationDisk. Formulation of the hydrogel was adopted from a previous ICG assessment study [40]. A black box was used to limit the influence of ambient light and therefore create a comparable situation to in vivo use of laparoscopes.

### 2.2. Laparoscopic Near-Infrared Fluorescence Systems

Four commercially available laparoscopic systems for clinical use were tested: (1) VISERA ELITE II, Olympus; (2) ENDOCAM Logic 4K platform (short: ENDOCAM), Richard Wolf; (3) IMAGE1 S™ 4U Rubina (short: Rubina), KARL STORZ; and (4) da Vinci Xi, Intuitive Surgical (Figure 1). The VISERA ELITE II has a xenon lamp and the da Vinci Xi a laser source for NIRF excitation, while the other two systems use light emitting diodes (LEDs). Gain, software-based, to increase the fluorescence intensity, was automatically adjusted to improve visibility, except for the manually adjustable VISERA ELITE II system. Instead of gain, brightness of fluorescence could be manually changed for the ENDOCAM, Rubina, and da Vinci Xi to improve visibility. All systems have (at least) two fluorescence view mode options, except the da Vinci system with only one fluorescence view mode. Description of components, indications, and fields of use can be found in Appendix A Table A2.

### 2.3. Measurements

Imaging of phantoms, with different concentrations of bevacizumab-800CW and ICG, was performed with all four laparoscopic NIRF systems. Since 15 dilutions were used, imaging was performed in two steps by using the phantom of Figure 1a; the highest eight concentrations were imaged together in Eppendorf tubes, followed by the lowest seven concentrations. First, standard system settings (brightness, intensity or gain) were compared to high settings. High settings meant that the brightness, intensity or gain were maximised. Furthermore, imaging was performed in the different fluorescence modes when available. Next, working distance between laparoscope and phantom were varied to represent general working distances in laparoscopic surgery (8.7, 9.6 and 10.4 cm) while keeping all ROIs in the field of view. Last, visibility and tissue penetration were determined under optimal conditions per system. Up to three layers of tissue-mimicking hydrogels (0.8 cm in width each) were stacked to assess tissue penetration.

### 2.4. Data Analyses

The acquired fluorescence images were independently assessed for visibility by two researchers (DJS and AJS). Visibility of individual Eppendorf tubes was scored as visible or invisible and disagreement was resolved through discussion or a third researcher (ECJ). Consensus judgement of visibility was used for analyses since the interobserver agreement was almost perfect (kappa = 0.88) [41].

Segmentation of images was performed to evaluate the different image properties by three different parameters; target-to-background (or tumour-to-background, TBR; Equation (1)), signal-to-noise (SNR; Equation (2)), and contrast-to-noise (CNR; Equation (3)) ratio. The fluorescence signal was expressed in arbitrary units (AU). Regions of interest were defined as the windows in the CalibrationDisk (Appendix A Figure A1). TBR represents sensitivity to detect a fluorescent object from background and is frequently reported in clinical studies [16,23,42,43]. SNR is a measure comparing the signal, and therefore defines the sensitivity, and may be used to define the applicability of a tracer [43,44]. CNR is a measure based on contrast instead of the raw signal [44,45] Noise from background may influence detectability, thus CNR may be more representative in an environment with noise. System settings and working distance with the highest visibility were selected to assess the lowest visible concentration and tissue penetration depth. Analyses were performed with Matlab (2022b, The MathWorks, Portola Valley, CA, USA). No comparative tests were performed since this study explored the use of laparoscopes with bevacizumab-800CW instead of directly comparing equipment. Furthermore, photobleaching of IRDye800CW and ICG during measurements would lead to unfair comparison of laparoscopes [46,47,48,49,50,51].
TBR = mean signal of ROI/mean signal of background(1)
SNR = mean signal of ROI/standard deviation of signal in non-ROI (background)(2)
CNR = (mean signal of ROI − mean signal in non-ROI (background))/standard deviation of signal in non-ROI (background)(3)

## 3. Results

### 3.1. Influence of System Settings and NIRF Mode on Visibility

Serial dilutions of bevacizumab-800CW (concentrations ranging from 0.42 to 6800 nM) and ICG (concentrations ranging from 394 nM to 6460 µM) were used to assessed visibility with different system settings and NIRF modes, totalling fourteen different scenarios with four laparoscopes for each fluorophore. Adjustment of brightness did not improve visibility of bevacizumab-800CW and ICG for the da Vinci Xi and ENDOCAM (Figure 2). All concentrations of bevacizumab-800CW were invisible with the standard settings of the VISERA ELITE II, independent from NIRF mode, while many of the ICG concentrations were visible with the same setting. However, bevacizumab-800CW was visible after the VISERA ELITE II system settings were manually adjusted. Likewise, adjusting fluorescence intensity led to visibility of lower concentrations when using the Rubina. Changing the fluorescence mode to a monochromatic view led to improvement in visibility of fluorophores for the VISERA ELITE II and Rubina, but it did not improve visibility of bevacizumab-800CW or ICG for the ENDOCAM. High system settings and monochromatic view often showed improved visibility and outcomes analysis.

### 3.2. Influence of Working Distance on Visibility

High camera settings and monochromatic NIRF mode were used to assess the influence of working distance to the phantom (Figure 3). In general, lower concentrations of bevacizumab-800CW were visible with the Rubina and VISERA ELITE II after the working distance was reduced to 8.7 cm. However, working distance did not influence visibility of bevacizumab-800CW and ICG for the ENDOCAM and da Vinci Xi. The smallest working distance, 8.7 cm, was used for assessment of visibility of concentrations and tissue penetration of the fluorescent signal.

### 3.3. Visibility of Bevacizumab-800CW

The lowest visible concentrations of bevacizumab-800CW for the da Vinci Xi, ENDOCAM, VISERA ELITE II, and Rubina system were 850, 850, 213 and 13 nM (corresponding to 8, 8, 32 and 512 times diluted stock), respectively. In Figure 4 this is shown in grey.

The TBR, SNR, and CNR of the lowest visible concentration bevacizumab-800CW were: (1) VISERA ELITE II 2.90, 1.94, 1.27; (2) ENDOCAM 0.86, 2.52, −0.41; (3) Rubina 1.75, 2.53, 1.08; (4) da Vinci Xi 1.44, 1.44, 0.44, respectively (Figure 4). Primarily for the ENDOCAM and da Vinci Xi, there were invisible concentrations that had comparable or higher ratios than the last visible concentration.

TBR, SNR, and CNR were higher for the visible ICG concentrations compared to the bevacizumab-800CW concentrations, especially for the Rubina and da Vinci Xi (Figure 5e–h). However, the lowest visible concentration of ICG was not found with the serial dilutions used in this set up as the lowest available concentration of ICG was still visible.

The ratios found using the VISERA ELITE II (Figure 5a,b) decrease when imaging ICG and the settings are changed from standard to high, especially for the SNR and CNR. Both are influenced by noise which increased with the use of higher settings. Switching to the monochromatic view of the ENDOCAM resulted in higher ratios compared to the white light fluorescence overlay mode (Figure 5c,d). In the Rubina system (Figure 5e,f) changing the system settings did not substantially change the ratios. Looking at the results using the da Vinci Xi (Figure 5g,h) it is striking to see that the SNR and CNR are higher when increasing the settings.

### 3.4. Tissue Penetration of Bevacizumab-800CW and ICG

Visibility of bevacizumab-800CW and ICG decreased with the amount of tissue-mimicking material (Figure 6). High concentrations (range 0.4–6.8 µM) of bevacizumab-800CW were visible in all systems with at least 0.8 cm of tissue-mimicking material, except for the da Vinci Xi. The Rubina and VISERA ELITE II were able to detect 1.7–6.8 µM bevacizumab-800CW through 1.6 cm of gel. In contrast, ICG could be detected through a minimum of 1.6 cm of tissue-mimicking material, and Rubina managed to detect ICG through 2.4 cm of gel. Interestingly, ICG concentration 9 with the VISERA ELITE II, and concentration 9 and 10 of the ENDOCAM were invisible using two gel layers while earlier and later concentrations were visible.

## 4. Discussion

Image-guided surgery and NIRF could aid surgeons in intra-operative decision-making. This study explored the use of IRDye800CW, conjugated to bevacizumab, in vitro with four commercially available NIRF laparoscopic surgical systems intended for use with ICG. All four laparoscopes were able to detect bevacizumab-800CW in the highest concentrations, relevant for lymph node metastases detection after local administration, even though optimal absorption and emission wavelengths of IRDye800CW are different from ICG [13,38,52]. Depending on the system, bevacizumab-800CW could be detected through up to 1.6 cm of tissue-mimicking material. In comparison, ICG was still visible after the standard solution was >16,000 times diluted and achieved 1.6–2.4 cm tissue penetration in all systems. In general, a surgeon could reduce the working distance and adjust system settings (gain, sensitivity or brightness) in order to improve visibility of bevacizumab-800CW.

In our study, laparoscopes were able to detect bevacizumab-800CW down to a concentration of 13–850 nM. In contrast, preclinical and clinical open-air camera systems, can detect ~0.05–15 nM bevacizumab-800CW in ideal conditions [14]. Visibility of bevacizumab-800CW with the laparoscopes might be sufficient for clinical use depending on the application, since an estimated 6–65,000 nM is required to detect ICG in sentinel lymph nodes after local injection [53] and IRDye800CW has a similar or higher quantum yield than ICG [54,55,56]. In an animal study, locally administered ^99m^Tc-tilmanocept-800CW was successfully used for sentinel lymph node detection with a laparoscope [36]. Intravenously administered tumour-targeted tracers in micro-dosages will have a lower percentage of tracer uptake in metastatic lymph nodes; an estimated 0.3% of the injected dose [53]. In comparison, colonic adenomas contained an estimated 4.8–6.86 nM bevacizumab-800CW after 10–25 mg intravenous bevacizumab-800CW which should not be detectable with the laparoscopes [57]. Thus, the four laparoscopes might be insufficient to detect tumour-targeted bevacizumab-800CW in metastatic lymph nodes after intravenous injection or more tracer must be administered. Furthermore, promising results have already been achieved with tumour-targeted IRDye800CW for the detection of metastatic lymph nodes with open camera systems [58,59].

There is a wide range of conditions that influence visibility of tracers. For example, reduced working distance improved visibility in our study like it did in another study that assessed bevacizumab-800CW with an endoscope (Xenon light source with ICG filters and Image1S H3-Z FI camera from KARL STORZ) [60], due to the inverse-square law [61]. However, light reflections of tissue also increase with a decreased working distance and could give false information which may complicate decision making during surgery [60]. In the subjective assessment, ROIs were sometimes white instead of their intended NIRF green or blue colour (depending on the system) in the white light with overlay images. White ROIs were not considered to contain fluorescence, since white ROIs were due to light reflecting from tissue-mimicking gel or from oversaturation which may be caused due to high system settings. Therefore, some ICG concentrations were intermittently visible as seen in Figure 6 (VISERA ELITE II and ENDOCAM). Furthermore, ICG would only become visible after the stock was diluted >16 times. Invisibility of ICG in the highest concentrations is due to the aggregation of ICG molecules, which results in a substantial lower absorption wavelength (695 nm) [38] compared to the usual 800 nm absorption peak of ICG [13]. Quenching might also contribute to invisibility of ICG in >0.5 µg/mL concentrations [62] (roughly corresponding to 1024 times diluted stock in our study), but new research contradicts this claim [63].

In this phantom study, tissue penetration of bevacizumab-800CW and ICG was up to 1.6 cm and 1.6–2.4 cm through tissue-mimicking material, respectively. In earlier phantom studies, ICG could be detected through 3.6 mm of beeswax (optimal concentration after 200–2500 times diluted) [64], 6 mm of prostate-mimicking phantom (optimal concentration after 2000 times diluted) [65], and 2 cm of breast tissue-mimicking material [40]. Thus, fluorescence penetration varies in different tissues or phantoms due to optical properties, in addition to working distance and imaging properties [14]. Human or animal mesocolon could have been used for in vitro recreation of colon cancer sentinel lymph node detection. However, problems arose with the use of pig mesocolon as the tissue was heterogeneous and had an unstable shape, which hinders testing multiple surgical systems over multiple hours. The tissue-mimicking material used in this phantom study was homogenous and had similar optical absorption properties to human fat, thus allowing optimal comparison of multiple systems with the use of a calibration device [40]. Radioactive tracers are capable of achieving deeper tissue penetration compared to NIRF tracers. Utilisation of hybrid radioactive and NIRF targeted tracers, such as ^99m^Tc-tilmanocept-800CW [36] and ^99m^Tc-ICG [66,67,68], would enable surgeons to benefit from the deeper tissue penetration of radioactive tracers for targeting lesions located further away while also allowing for more precise localisation of lesions in close range using NIRF. Future incorporation of augmented reality based on preoperative imaging presents another potential application for these hybrid tracers [69].

The lowest visible concentrations bevacizumab-800CW had a TBR ranging from 0.86 to 2.90, a SNR of 1.44 to 2.53, and a CNR of −0.41 to 1.27. Literature suggests that a TBR > 1.5 might be sufficient to discriminate fluorescent lesions and reduce false negatives in ideal situations. However, often a TBR > 2 is pursued for fluorescence-guided surgery and decision-making, and the TBR is also dependent on where it is measured [70]. For example, in vivo TBR would probably result in lower TBRs compared to ex vivo or in vitro measurements [70]. CNR might be more representative in an environment with noise and a CNR < 0 means less signal than noise, meaning that the ROI is not distinguishable from the background noise [44]. However, clear cut-off values for the ratios are not defined in the literature. Although visibility is a subjective parameter, this study showed a near-perfect interobserver agreement and its subsequent clinical applicability. Furthermore, objective image quality (described by SNR and CNR) did not always correspond to the subjective clinical visibility score. Some of the lowest visible concentrations were judged as visible but had a CNR < 0 and a TBR < 1. This could be due to multiple reasons. First, in the lowest visible samples the fluorescence signal was scarcely visible within the ROI, while the larger part of the remaining ROI was dark. This reduces the mean value of the ROI compared to the background leading to a negative CNR and low TBR. Second, the conversion of colour images to a grey scale, for segmentation purposes, may have decreased contrast. Two images, one acquired in monochromatic mode and one in coloured NIRF overlay mode, would result in different TBR/SNR/CNR after this segmentation process, but would have identical (subjective) visibility. Last, disagreement between objective and subjective assessment was seen in invisible concentrations that had a higher TBR/SNR/CNR than the last visible concentration, predominantly occurring with the ENDOCAM and da Vinci Xi. This might be explained by the automatic gain that is automatically applied to dark images in which no (or very little) fluorescence is detected. Often the second set of bevacizumab-800CW serial dilutions (with the lowest seven concentrations) in the phantom were predominantly invisible. Therefore, the second set of images were darker compared to first set of eight concentrations and more automatic gain would be applied.

### Strengths and Limitations

To our knowledge, this is the first study looking into the use of commercially available laparoscopes of key market players and the use of a tumour-targeted tracer without modification of hardware or software. For this study, IRDye800CW conjugated to only bevacizumab was used as a tumour-targeted tracer. However, we hypothesise that other antibodies conjugated to IRDye800CW will have similar results, since the same NHS ester group is used for conjugation. Furthermore, absorption and emission wavelengths of IRDye800CW and hybrid targeted IRDye800CW tracers (e.g., ^111^In-DTPA-trastuzumab-800CW and ^68^Ga-tilmanocept-800CW) were comparable [71,72], unlike free ICG and ICG bound to albumin [38]. However, antibody-bound IRDye800CW or other tumour-targeted tracers are currently not yet approved by the FDA or EMA for application in general surgery. Nevertheless, the market for fluorescence-guided surgery is growing which might stimulate development of laparoscopic NIRF systems for tumour-targeted tracers [14]. A high sensitivity of laparoscopic NIRF systems for a tracer is one of the six key features described by Dsouza et al. and is especially important for tracers in micro-dosages [14]. However, micro-dosages of tumour-targeted bevacizumab-800CW have already been used for a broad range of indications with (experimental) open-air and endoscopic systems [18,19,21,24,57,59,73,74,75,76]. Thus, translation to optimised laparoscopes should be achievable. In addition, the intended use for the laparoscopes, except the Olympus (Appendix A Table A2), already allows the use of fluorescence imaging for other fluorophores than ICG (e.g., IRDye800CW).

The different dilutions used, were too many to image all at once using the CalibrationDisk. Therefore, the highest eight concentrations and the lowest seven concentrations were imaged separately. This may have resulted in an unexpected intensity increase or decrease at the middle dilutions due to automatic adjustment of the images by some of the systems as also earlier described to cause an objective and subjective disagreement. Ideally a control dilution, used in both images, should have been used or all the used dilutions should have fitted in the CalibrationDisk.

Furthermore, the Rubina and da Vinci Xi laparoscopes had 30° optics while the other two laparoscopes were 0°. Ideally, only 0° optics would have been used, but the Rubina and da Vinci Xi 0° optics were unavailable at the time of the experiments. Measurements of the Rubina and da Vinci Xi were not postponed to keep all other variables constant. Nonetheless, the central viewing axis and working distances were kept constant between the different degree optics as these variables influence fluorescence intensity [64]. Furthermore, we do not expect the degrees of the optics to influence results as evident from a prior study [60]. However, we did not control for photobleaching of bevacizumab-800CW and ICG as we used the same samples for every laparoscope. Fluorescence intensity can halve after 30 min of near-infrared exposure [49]. Therefore, no conclusions can be drawn about which laparoscope is most sensitive to the fluorophores and future studies may want to control for photobleaching [40]. However, this study does suggest there are differences between usability of laparoscopes due to the wide range of the lowest visible concentration bevacizumab-800CW (13–850 nM). Consequently, surgeons should be aware that studies with fluorescent tracers (i.e., IRDye800CW and ICG) cannot directly be extrapolated to other laparoscopes or fluorescence imaging systems. Above all, each system has its own negative and positive properties and users should be aware of those properties when choosing a laparoscope for fluorescence imaging. Until there is a clear standardisation protocol for fluorescence imaging it will remain difficult to compare the sensitivity of different systems [77].

## 5. Conclusions

IRDye800CW, conjugated to bevacizumab, was visible with four commercially available laparoscopic surgical systems optimised for ICG with the lowest visible concentrations being 13–850 nM (equal to 8–512 times diluted stock). Tissue penetration of bevacizumab-800CW was up to 1.6 cm. In contrast, ICG was still visible after >16,000 times dilution and achieved tissue-penetration up to 2.4 cm. Depending on the application, visibility of bevacizumab-800CW might be sufficient for clinical use with current NIRF laparoscopes, but optimisation of sensitivity for IRDye800CW would widen the indication of tumour-targeted tracers in laparoscopic surgery. More future studies should focus on standardised testing of laparoscopes and their capabilities for detecting (new) NIRF tracers.

## Figures and Tables

**Figure 1 diagnostics-13-01591-f001:**
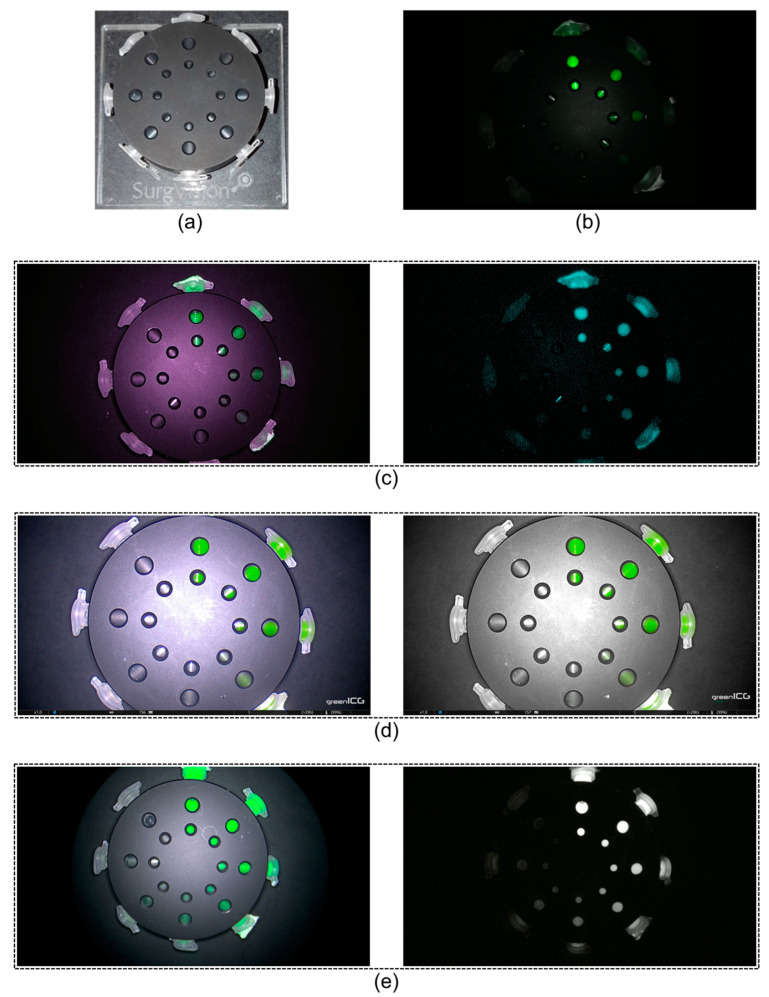
CalibrationDisk and near-infrared fluorescence modes of laparoscopes. (**a**) The CalibrationDisk for near-infrared fluorescence calibration with eight transparent Eppendorf tubes, and (**b**–**e**) serially diluted bevacizumab-800CW visible with four laparoscopes. (**b**) The da Vinci Xi had a single, monochromatic, fluorescence mode. (**c**) The VISERA ELITE II, (**d**) Rubina, and (**e**) ENDOCAM had a white light with fluorescence overlay mode (left column) in addition to a monochromatic mode (right column). The Rubina has a third mode, intensity map, that is not displayed since it was not tested in this study. The fluorescent light emitted by the fluorescent dyes were displayed as (a false) green or blue colour and superimposed over the images, except for the Rubina monochromatic image which was a true monochromatic mode. The pictures were all acquired at the same working distance (10.4 cm), therefore, differences in field of views depend on system properties.

**Figure 2 diagnostics-13-01591-f002:**
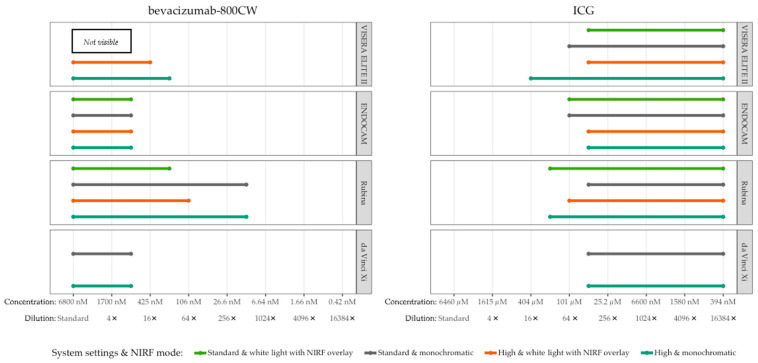
Visible concentrations of bevacizumab-800CW and ICG with four laparoscopes using varying system settings and NIRF modes. Adjusting system settings and switching to a monochromatic NIRF mode generally improved visibility of fluorophores for the VISERA ELITE II and Rubina. For example, bevacizumab-800CW was not visible with the standard settings of the VISERA ELITE II while it was visible with high system settings. Note(s): the da Vinci Xi does not have a white light with NIRF overlay mode.

**Figure 3 diagnostics-13-01591-f003:**
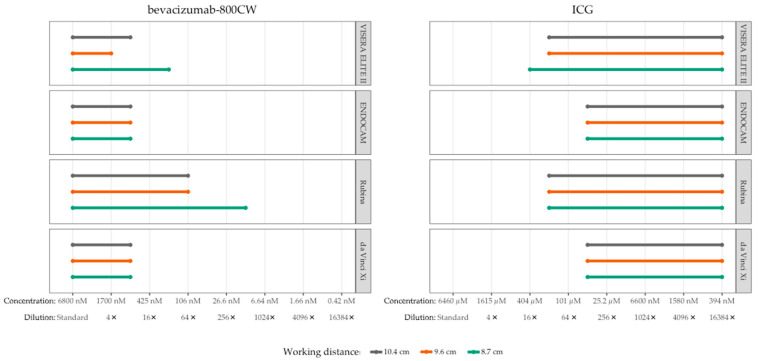
Visible concentrations of bevacizumab-800CW and ICG with four laparoscopes using varying working distances (8.7, 9.6 and 10.4 cm). Working distances were measured from tip of the laparoscope to the phantom. More concentrations of fluorophores were visible with reduced working distance using the VISERA ELITE II and Rubina, in accordance with the inverse-square law. The working distance did not influence visibility for the ENDOCAM and da Vinci Xi. Note(s): measurements performed with high system settings and monochromatic NIRF mode.

**Figure 4 diagnostics-13-01591-f004:**
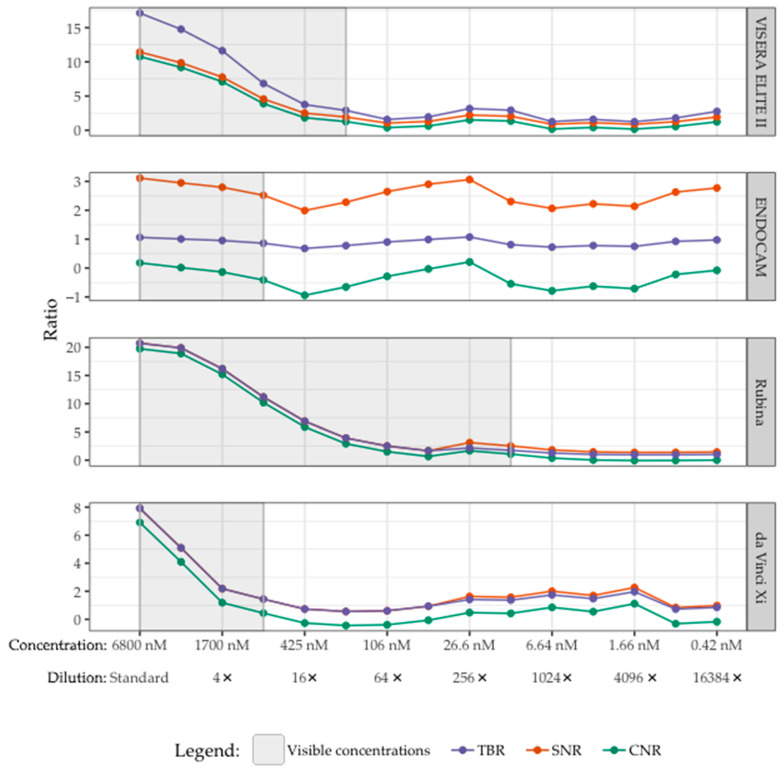
Visibility of bevacizumab-800CW with four laparoscopes correlated with TBR, SNR, and CNR. The visible concentrations per laparoscope are displayed in grey. The lowest visible concentrations of bevacizumab-800CW were 850, 850, 213 and 13 nM (corresponding to 8, 8, 32 and 512 times diluted stock) with the da Vinci Xi, ENDOCAM, VISERA ELITE II and Rubina, respectively. The TBR, SNR, and CNR decreased until bevacizumab-800CW was not visible anymore. However, there were higher TBRs/SNRs/CNRs for some invisible concentrations compared to the last visible concentration of bevacizumab-800CW, especially with the ENDOCAM and da Vinci Xi. Note(s): measurements performed with high system settings, monochromatic NIRF mode, and 8.7 cm working distance.

**Figure 5 diagnostics-13-01591-f005:**
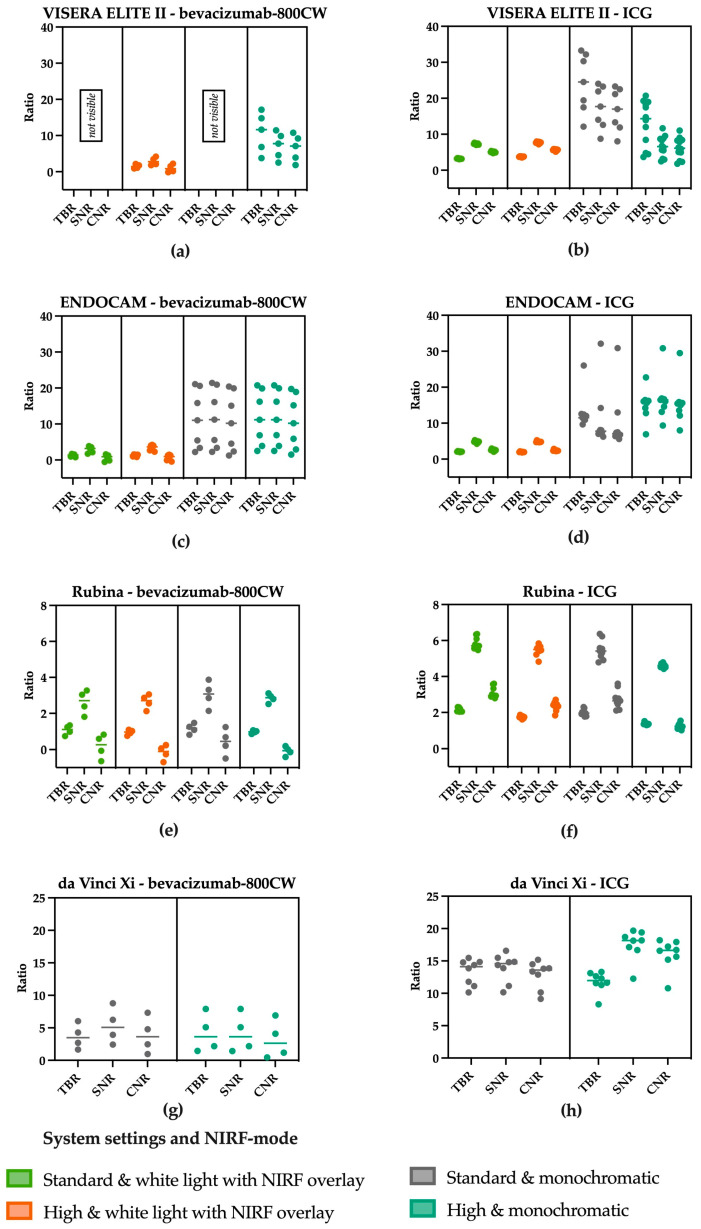
TBR, SNR and CNR of the visible concentration of bevacizumab-800CW and ICG using varying settings on laparoscopes. (**a**) bevacizumab-800CW using the VISERA ELITE II. (**b**) ICG using the VISERA ELITE II. A change in settings from standard to high increased the noise and therewith decreased the SNR and the CNR using the VISERA ELITE II. (**c**) bevacizumab-800CW using the ENDOCAM. (**d**) ICG using the ENDOCAM. Higher ratios were seen using the monochromatic view, suggesting better visibility using the ENDOCAM. (**e**) bevacizumab-800CW using the Rubina system. (**f**) ICG using the Rubina system. The Rubina barely showed changes in the ratios when changing the settings. (**g**) bevacizumab-800CW using the da Vinci Xi. (**h**) ICG using the da Vinci Xi. When increasing the system settings in the da Vinci Xi, a slight increase in the ratios was noted. Note: measurements performed with 8.7 cm working distance.

**Figure 6 diagnostics-13-01591-f006:**
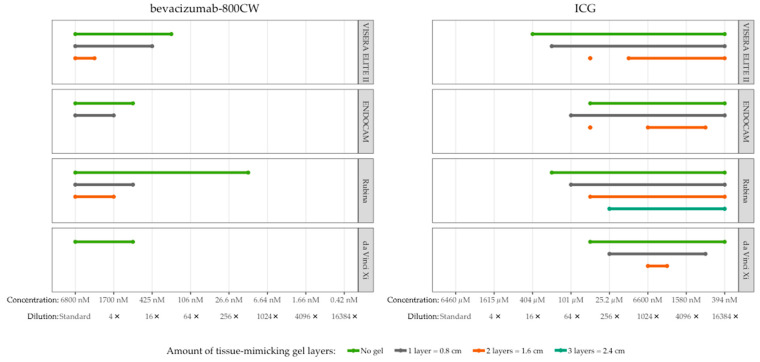
Tissue penetration of bevacizumab-800CW and ICG with four laparoscopes using tissue-mimicking gel. Bevacizumab-800CW was visible with all laparoscopes through 0.8 cm of gel, except for the da Vinci Xi (not visible through gel), and the Rubina was able to detect fluorescence through 1.6 cm of gel. By contrast, ICG could be detected through a minimum of 1.6 cm of gel, up to 2.4 cm with the Rubina. Note(s): measurements performed with high system settings, monochromatic NIRF mode, and 8.7 cm working distance.

## Data Availability

The data presented in this study are available on reasonable request from the corresponding author.

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
