# Peer review of "Detection of Tumour-Targeted IRDye800CW Tracer with Commercially Available Laparoscopic Surgical Systems"

_diagnostics, 2023, doi:10.3390/diagnostics13091591_

Round 1

Reviewer 1 Report

In my opinion it’s well structured, easily understandable and overall just very well written.

It would have been interesting to see the results of the study if the experiments were conducted in a human, but I understand that that’s a subject for another study in the future 

All the graphics are clear and relevant.

It’s a very interesting topic and I can’t wait to follow its evolution and possible impact on the medical world because I think that this could change the surgical approaches. 

I really hope that the problems the researches described can and will be improved.

Nothing to add, article is easy to read, formulated properly and easy to understand. 

Author Response

Response to the reviewer:
Thank you for your overwhelmingly positive feedback, for expressing interest in our research topic, and
recommending our manuscript for acceptance for publication. We appreciate the time and effort you took to
review our work and provide valuable insights. Your comments and suggestions have helped us improve the
quality of our manuscript, particularly about the Internet of Surgical Things (IoST). We think that augmented
reality could be beneficial in the operating room and it is a topic that we are actively researching. For this reason,
we have added this topic to the discussion of our manuscript and used your suggested article as a reference. We
are grateful for your support.

Reviewer 2 Report

I was glad to review the work of the authors regarding this very interesting article entitled "Detection of tumour-targeted IRDye800CW tracer with commercially available laparoscopic surgical systems ". The manuscript is well-written and the incorporated table and figures make the study easy to follow.

I strongly recommend acceptance for publication of the paper after minor changes.

Despite the major advances in fluorescence-guided surgery, there are still numerous unanswered questions regarding near-infrared fluorescence (NIRF) and its’ new applications for fluorescence-guided surgery.

“In the last few years, technological developments in the surgical field have been rapid and are continuously evolving. One of the most revolutionizing breakthroughs was the introduction of the IoT concept within the surgical practice.”

Add this information in the discussion section and explain the role of IoT in Image-guided surgery, such as near-infrared fluorescence (NIRF) in surgical decision-making.

Consider citing the article on Internet of surgical things

https://pubmed.ncbi.nlm.nih.gov/35746359/

Author Response

(The authors gave the same response as above.)

Reviewer 3 Report

Review Report

The authors have presented a paper on the investigation of the use of 20 IRDye800CW, conjugated to bevacizumab, with four commercially available NIRF laparoscopes. The paper is well written and contains enough results to be considered. However, I would like to suggest the following minor corrections before the acceptance of the manuscript for publication in the Diagnostic Journal.

1.     The authors need to review the infrared range in line 51.

2.     Please keep space between text and square bracket for citations throughout the manuscript

3.     Please increase the size of the letters in Figure 1

4.     Line 391, please revisit the positioning of the heading and numbering. Also, I suggest extracting part of it to improve the conclusion which too short and needs further improvement.

Author Response

Response to the reviewer:
Thank you for your overwhelmingly positive feedback, for expressing interest in our research topic, and
recommending our manuscript for acceptance for publication. We appreciate the time and effort you took to
review our work and provide valuable insights. Your comments and suggestions have helped us improve the
quality of our manuscript. To respond to your comments point-by-point:

1.
We changed to sentence on line 51 to more accurately reflect our earlier statement to the following:
Near-infrared light (650-950 nm for the first NIRF window) has superior tissue penetration compared
to white light and has limited interference with human tissues.”

2. We added space between text and square brackets throughout the manuscript

3. We increased the font size in Figure 1.

4. We made the suggested change to line 391.
Our intention was to keep the conclusion short and concise, with our results forming the basis. In
response to feedback, we have expanded the conclusion to include, among other things, what future
research should focus on, as discussed in the strengths and limitations.